# Cloud-Based Single-Frequency Snapshot RTK Positioning

**DOI:** 10.3390/s21113688

**Published:** 2021-05-26

**Authors:** Xiao Liu, Miguel Ángel Ribot, Adrià Gusi-Amigó, Adria Rovira-Garcia, Jaume Sanz, Pau Closas

**Affiliations:** 1Albora Technologies, London SE1 6LN, UK; miguel@albora.io (M.Á.R.); adria@albora.io (A.G.-A.); closas@ece.neu.edu (P.C.); 2Research Group of Astronomy and GEomatics (gAGE), Universitat Politecnica de Catalunya (UPC), 08034 Barcelona, Spain; adria.rovira@upc.edu (A.R.-G.); jaume.sanz@upc.edu (J.S.); 3Electrical and Computer Engineering Department, Northeastern University, Boston, MA 02115, USA

**Keywords:** snapshot RTK, cloud-based positioning, coarse time navigation, satellite-based navigation, precise positioning, integer ambiguity resolution

## Abstract

With great potential for being applied to Internet of Things (IoT) applications, the concept of cloud-based Snapshot Real Time Kinematics (SRTK) was proposed and its feasibility under zero-baseline configuration was confirmed recently by the authors. This article first introduces the general workflow of the SRTK engine, as well as a discussion on the challenges of achieving an SRTK fix using actual snapshot data. This work also describes a novel solution to ensure a nanosecond level absolute timing accuracy in order to compute highly precise satellite coordinates, which is required for SRTK. Parameters such as signal bandwidth, integration time and baseline distances have an impact on the SRTK performance. To characterize this impact, different combinations of these settings are analyzed through experimental tests. The results show that the use of higher signal bandwidths and longer integration times result in higher SRTK fix rates, while the more significant impact on the performance comes from the baseline distance. The results also show that the SRTK fix rate can reach more than 93% by using snapshots with a data size as small as 255 kB. The positioning accuracy is at centimeter level when phase ambiguities are resolved at a baseline distance less or equal to 15 km.

## 1. Introduction

In recent years, applications of Internet of Things (IoT) and Location Based Service (LBS) have gained much greater attention from the industry [1] and research communities [2,3,4]. Behind these popular concepts are the fast development of technologies such as Global Navigation Satellite Systems (GNSS) [5], cloud computing and 4G/5G communication. The great pace of these developments brought a number of enabling capabilities; we mention here three main aspects related to the work in this article. First of all, mobile communication modules allow r aw data to be transmitted from the sensor front-end to a remote data center for further processing. The fast pace of the 4G/5G base station deployment has made internet access available for much larger areas with faster speed and lower delay. Secondly, an increasing number of cloud computing platforms have been developed by different service providers; better computation power and greater data storage capabilities are available to individuals or companies with easy access and great flexibility. Finally, an ever increasing number of GNSS reference stations are being deployed worldwide and reference correction data distributed ensuring shorter baseline distances to the GNSS user receivers, which is critical for Real Time Kinematics (RTK) applications [6].

With all the benefits brought by these recent developments, cloud-based snapshot positioning has been proposed aiming at IoT applications such as asset tracking and logistics. The basic work flow was introduced [7] and primary results were evaluated in many papers [8,9,10]. The cloud-based snapshot positioning technique collects a short period of GNSS signal, named as a snapshot, and places the subsequent signal and data processing procedures on the cloud in order to take full advantage of the previously mentioned benefits of cloud computing. This technique has improved the efficiency substantially in terms of power consumption compared to the conventional receivers since the majority of computation burdens are moved to the remote server side. Furthermore, snapshot receivers can enable a duty cycle mechanism to further reduce the power consumed on the user side, that is, switch off the receiver or shift to a sleep mode when there are no positioning needs. On the contrary, conventional receivers have to keep running on board until valid GNSS measurements can be generated before obtaining a RTK solution. This advantage is extremely useful for certain applications such as asset tracking that requires position information only a few times a day. Moreover, this technique also takes full advantage of the high computational capability of the cloud computing infrastructures, so that the complex tasks such as signal acquisition can be performed faster and the time to obtain a solution can be reduced. These advantages of cloud-based snapshot positioning come along with some drawbacks, mainly the need of a communication module for sending snapshots to the cloud and the possible latency of positioning results caused by the data transmission and signal processing delay on the cloud.

While the advantages of cloud-based snapshot positioning in power efficiency and computational capabilities have been well addressed by the GNSS industry [1,8], its positioning accuracy has not yet received adequate attention. In order to improve the positioning accuracy, the authors have developed a new GNSS positioning technique on the basis of cloud-based snapshot positioning, which is called Snapshot RTK (SRTK). In [11], the authors proved theoretically that SRTK is feasible under certain Signal-to-Noise-Ratio (SNR) conditions after experimenting on simulated data. Based on that, we have recently confirmed the feasibility of SRTK using GNSS signals collected from the real world, although our experiments were limited to zero-baseline configurations only [12]. The present article extends this previous work and provides a deeper insight into the strategies used in the measurement generation process. It also evaluates the performance of SRTK in terms of Integer Ambiguity Resolution (IAR) and final positioning accuracy. Particularly, the experiments described in this paper consider different baseline distances of up to 50 km instead of the zero-baseline case considered before in [12]. In addition, since collecting snapshot data from multiple frequency bands will increase the total snapshot size, only single frequency data are used in the present research, confirming that single frequency processing is sufficient to obtain a high fix rate for most of the scenarios discussed in Section 3. With SRTK, applications requiring high-precision, such as Connected and Automated Vehicles (CAV) [13], geodetic measuring, or deformation monitoring of critical infrastructures, can benefit from snapshot-based positioning as well.

From a commercial point of view, one of the major costs of adopting the SRTK service comes from the transmission of raw GNSS data from the user end to the remote cloud platform. The mobile network data plan chosen for this service will depend on the minimum size of the data packet needed for each snapshot. The smaller the snapshot, the cheaper this service can be. This work mainly aims to explore the minimum data size of a snapshot required for obtaining a given IAR fix rate. The snapshot size depends on the data quantization, the sampling rate, and duration of the data collection configured in the GNSS front-end. The sampling rate directly influences the maximum bandwidth of the signal, and the duration of the data limits the total integration time available for satellite acquisition. Those constraints further influence the noise level of generated GNSS observables, which in the end determines the IAR fix rates. Furthermore, the baseline distance between the user end and the GNSS reference station also largely impacts the feasibility of achieving IAR. Different experiments are organized in this paper to explore the performance differences brought by these factors.

From a technical perspective, this work mainly addresses innovations and improvements in the following three aspects: The first contribution of this work is the development of a novel algorithm that ensures a nanosecond level absolute timing accuracy so that satellite positions can be computed with a much higher accuracy as well.The present research also addresses the problems brought by the lack of knowledge of the encoded navigation data bits, noted as the data bit ambiguity issue in this manuscript. This issue impacts the fractional carrier phase measurements and determines the possibility of obtaining an SRTK fixed solution, a method is proposed in this work to tackle this issue.The two previous innovations, as well as an improved data processing workflow of the novel SRTK algorithm, builds on top of our previous work [12]. To the best of our knowledge, this is the first time that snapshot signals have been used to generate RTK fixed solutions under a non-zero baseline configuration.

The present research highlights the following two benefits for the GNSS industry. On the one hand, it provides a clear view of the relationships between the snapshot receiver settings and the expected positioning performance, which could be critical when deciding which parameters to use in order to reduce size of snapshot data and its transmission cost. On the other hand, the baseline distance analysis also brings useful insights for service providers, especially about the GNSS base station network density before the deployment.

## 2. Methodology

This Section first introduces the general workflow of the proposed cloud-based SRTK algorithm. Then the main steps of this process and the methods used to tackle the existing challenges are described in detail. Finally, the three key parameters that influence the RTK fix rate, i.e., snapshot signal bandwidth, integration time, and baseline distances, are discussed.

### 2.1. Snapshot RTK Processing Workflow

The general workflow of cloud-based GNSS positioning was described in previous works [9] and this architecture was further investigated and adapted to achieve high-precision positioning in [12]. Figure 1 illustrates the basic workflow considered in this article.

GNSS signals are received by the antenna and then processed by the snapshot receiver, which is similar to the front-end part of a conventional receiver that mainly performs the tasks of converting the RF signal to a stream of digital bits and down-converting the signal to baseband or an Intermediate Frequency (IF). The resulting output file is the snapshot data that will be transmitted to the cloud platform by the cellular network module. This data set contains 2 streams of digital bits; one corresponds to the in-phase samples and the other corresponds to the quadrature samples. The size of this particular file is a main focus of this paper. After the snapshot samples arrive at the cloud server, they are further processed in an SRTK engine together with other assistance data collected through different cloud services in order to estimate the PVT solutions of the snapshot receiver. Such assistance data include rough coordinates of the receiver that may be provided by the mobile network module, approximate time information that can be accessed through internet, reference measurements collected from a nearby GNSS base station and navigation data that are distributed by multiple organizations nowadays.

The specific steps inside the implemented SRTK engine are shown in Figure 2. The snapshot samples first go through an assisted acquisition process that generates the primary results of code delay, Doppler frequency offset and carrier phase for each acquired satellite. Note that before this step the number of candidate satellites to acquire have been reduced by a filtering process based on the assistance data. From here, the SRTK engine workflow can be divided into two paths. The upper panel, shaded in red color, is a typical Coarse Time Navigation (CTN) filter that only provides a Single Point Positioning (SPP) solution with meter level accuracy, more details about its implementation can be found in numerous studies [14,15,16,17]. The second path, shaded in green color, represents the unique implementations of SRTK. It includes a measurement generation step combined with a typical RTK navigation filter. With the help of reference measurements collected from a nearby station, a much more precise RTK PVT solution is obtained. When no reference data are available, the SRTK is disabled and only SPP outputs will be generated. Results from the coarse time SPP filter contain useful Time Of Week (TOW) information that is needed for the measurement generation process of SRTK; thus, the upper path is always computed first and a valid SPP PVT solution is necessary for the second path of SRTK workflow.

### 2.2. Acquisition

Currently, there are mainly two receiver design choices in order to obtain valid measurements of code delay, Doppler offset and carrier phase. This brings 2 types of GNSS positioning architecture: the closed-loop sequential tracking architecture and the open-loop batch processing architecture [7]. The former is used in most of the GNSS receivers currently in the market. It computes the code delay errors using, for example, an early-minus-late discriminator and the carrier phase errors using a Costas loop. These measurement errors are then utilized in a feedback loop to generate a local replica that better aligns the received signal. This architecture provides a continuous measurement output coming from these replicas. The latter, one the contrary, computes the correlation between the input signal and a batch of replica signals that are constructed based on the code and frequency search spaces, then the outputs are generated by seeking for the parameters that lead to the maximum energy. This method is less commonly used but is more practical for snapshot data due to the short duration of the recording and, at the same time, it can improve the positioning robustness in harsh signal reception scenarios [18]. The open loop strategy was also chosen for this research.

The main goal of the acquisition module is to estimate the code delay, Doppler offset and carrier phase for each satellite signal. The code delay measurements of each satellite in view are computed by exploiting the autocorrelation properties of the Pseudo-Random Noise (PRN) codes [5,19]. The coarse position and time information in the assistance data is used to obtain a rough estimate of the Doppler values; based on that, a few initial guesses of Doppler offset are used together with the known PRN sequences in order to form a batch of local replicas. After that, a Cross Ambiguity Function (CAF) is computed and the primary results can be obtained by searching for the parameters that lead to the maximum CAF magnitude. The CAF is a complex-valued function that is modeled as [20]:(1)$$Y\left(\tau ,F_{D}\right)\;=Y_{I}\left(\tau ,F_{D}\right)\;+\;jY_{Q}\left(\tau ,F_{D}\right),$$
where $Y_{I}\left(\tau ,F_{D}\right)$ and $Y_{Q}\left(\tau ,F_{D}\right)$ are the in-phase and quadrature components of the CAF and represented, respectively, as follows:(2)$$\begin{array}{c} Y_{I}\left(\tau ,F_{D}\right)\;=\frac{1}{N}\sum_{n=0}^{N-1}r\left[n\right]\cos\left(2\pi F_{D}n\right)c\left[n-\tau \right],\\ Y_{Q}\left(\tau ,F_{D}\right)\;=\frac{1}{N}\sum_{n=0}^{N-1}\;-\;r\left[n\right]\sin\left(2\pi F_{D}n\right)c\left[n-\tau \right]. \end{array}$$

Here, τ and $F_{D}$ stand for the code delay and Doppler frequency normalized by the sampling rate $f_{s}$, i.e., $F_{D}=f_{d}T_{s}=\frac{f_{d}}{f_{s}}$ with $T_{s}$ representing the sample duration. r[n] is the received signal and c[n−τ] is the shifted local replica of the PRN code. *N* is the number of samples, which is determined by the duration of coherent integration time used in acquisition and the sampling rate selected.

Note that the previous equations have ignored the effects brought by the possible phase changes of the navigation bits or secondary codes. In closed-loop architecture, a bit synchronization process is usually implemented to handle this issue, and the navigation bits are demodulated and wiped out in the following tracking loops. However, when computing the CAFs for open loop architectures, it is important to introduce multiple hypotheses for these data bits in order not to degrade the correlation peak magnitude [21]. For example, GPS L1CA signals are modulated with navigation data bits with a duration of 20 ms. If the coherent integration time is set to 20 ms as well, there could be one navigation bit sign change at any of the code period edges inside this signal recording. Therefore, 20 hypotheses are taken, corresponding to the possible bit shifts, and when there is a bit transition, the correct shift will maximize the CAF correlation peak. Note that these hypotheses need to be made for each satellite and are treated independently. Similarly, different hypotheses need to be made for other signals as well depending on their specific code structures. Thus, the CAF needs to be modified taking into account these data bit hypotheses. For the example of GPS L1CA signal with 20 ms integration time, the CAF can be represented as follows, assuming a data bit transition happening after the *u*-th code period:(3)$$\begin{array}{cc} Y_{I}\left(\tau ,F_{D},u\right) & =\sum_{v=1}^{20}H_{v}^{\left(u\right)}\cdot Y_{Iv}\left(\tau ,F_{D}\right),\\ Y_{Q}\left(\tau ,F_{D},u\right) & =\sum_{v=1}^{20}H_{v}^{\left(u\right)}\cdot Y_{Qv}\left(\tau ,F_{D}\right). \end{array}$$

This equation includes the parameter $H_{v}^{\left(u\right)}$ that represents the assigned bits for the *v*-th code period under the *u*-th hypothesis, where *u* is a value between 1 and the total number of hypotheses $N_{h}$. This value is defined as [18]:(4)$$H_{v}^{\left(u\right)}=\;\left\{\begin{array}{cc} 1, & v\leq u\\ -1, & v>u \end{array}.\right.$$

Note that the expression of the assigned data bits $H_{v}^{\left(u\right)}$ can become more complicated when longer integration times are considered. This is due to the fact that multiple bit transitions can, and likely will, take place for integration times longer than a single bit period. The total number of hypotheses $N_{h}$ will also grow in such scenarios.

By searching for the CAF’s maximum magnitude, the code delay, Doppler offset and navigation bit hypothesis can be determined. In addition, an interpolation step needs to be implemented for both the code delay and Doppler offset in order to obtain sufficiently precise measurements [7] for the SRTK navigation filter; this results in the final estimates of code delay ($\hat{\tau }$) and Doppler-shift (${\hat{F}}_{D}$).

In addition, the carrier phase at the beginning of the collected signal can be computed by the following equation based on these code delay and Doppler offset estimates:(5)$$\hat{\varphi }=\arctan\left(\frac{Y_{Q}\left(\hat{\tau },{\hat{F}}_{D},u\right)}{Y_{I}\left(\hat{\tau },{\hat{F}}_{D},u\right)}\right).$$

### 2.3. Measurement Generation

From the previous section we can see that the acquisition outputs actually include four values for each satellite: code delay $\hat{\tau }$, Doppler offset ${\hat{F}}_{D}$, carrier phase value $\hat{\varphi }$ and the the correct data bit or secondary code shift hypothesis *u*. There are 3 challenges to be addressed before these results can be used in the SRTK navigation filter. Namely, (1) the code delays are fractional values. That is to say, their values are always between 0 and 1 of the satellite signal’s primary code period, thus a set of full pseudoranges need to be computed by augmenting these fractional values with a proper number of full code periods. Fortunately, this can be achieved thanks to the assistance data as explained in Section 2.3.1; (2) the carrier phase measurements are also fractional values between the 0 and 1 cycle, although this is not a problem for the RTK navigation filter since a single-epoch IAR is to be performed. However, the real challenge for carrier phase measurements is caused by the fact that the so called “data bit ambiguities” can not be determined for all satellites in a consistent way. It is necessary to compensate for a half cycle bias for some satellites in order to ensure a constant common phase bias that could yield a valid RTK fix. This problem can be tackled by making use of data sources distributed by external organizations [22]; and (3) the timing accuracy plays an important role. For high precision applications like RTK, it is important to know exactly when the signals are transmitted in order to have satellite positions computed correctly. The following subsections give more details about the methodologies used to address these challenges.

#### 2.3.1. Full Pseudorange Generation

As mentioned before, it is necessary to assign a correct integer number of code periods to each satellite in order to build a valid geometric range relationship between satellites. This process is named as pseudorange alignment by the authors, and the integer numbers found for these code delays are called code ambiguities in this paper, as a counterpart of carrier phase ambiguities. In principle, the full pseudorange of the *k*-th satellite can be represented as:(6)$$P^{\left(k\right)}=c\cdot \left({\hat{\tau }}^{\left(k\right)}+N_{c}^{\left(k\right)}\cdot T^{\left(k\right)}\right),$$
where:*P* represents the full pseudorange value, in meters;$\hat{\tau }$ is the fractional code delay value obtained from the acquisition step, in seconds;$N_{c}$ stands for the code ambiguity, which is an integer value;*T* is the duration of a primary code periods, in seconds. For example, for GPS L1CA signal this value is 0.001 s and for Galileo E1C signal it equals to 0.004 s;*c* stands for the speed of light constant, in m/s.

The next step is to make sure that the geometric range values are consistent among all satellites. This can be verified with the help of the coarse time and position information. We define the following parameter $n_{c}^{\left(k\right)}$ as the float code ambiguity
(7)$$n_{c}^{\left(k\right)}=\frac{1}{c}∥r^{\left(k\right)}-r_{0}∥-{\hat{\tau }}^{\left(k\right)},$$
where $r^{\left(k\right)}$ is the position of satellite *k* computed at the given coarse time $t_{0}$ and $r_{0}$ is the rough position of the receiver. The flight time differences among satellites due to their different positions at the time of signal reception has been removed in this equation. However, there is still a common bias remaining in these values that need to be cleared. For this reason, we select the satellite with smallest float code ambiguity as a reference satellite and define it as $n_{c}^{\left(0\right)}$. Then an integer can be found for each of the remaining satellites by the equation
(8)$$N_{c}^{\left(k\right)}=\mathrm{round}\left(\frac{n_{c}^{\left(k\right)}-n_{c}^{\left(0\right)}}{T^{\left(k\right)}}\right),$$
where $T^{\left(k\right)}$ is the duration of the primary code period for satellite *k*. Different signals have different primary code period lengths; to simplify this issue, usually the 1-ms code ambiguities are computed by fixing $T^{\left(k\right)}=10^{-3}$ s, since 1 ms is a common divisor for the primary code duration of all GNSS signals. This is usually called the millisecond-ambiguity issue in other literature [14,15].

Now we have found a proper integer value that can be assigned to the fractional code delays and assure that they are well aligned to the signal reception time. Note that a constant integer, for example 68, can be added in order to adjust their values to be close to the nominal flight times of GNSS signals; this addition is optional as the existing common bias part will be assimilated to the receiver clock bias term in the navigation filter and does not influence SPP positioning accuracy.

Notice that there is a rounding process in this equation; this means that, if the errors brought by the inaccuracy of the coarse position and time are too large, a wrong integer maybe computed. More specifically, the combined effect of these assistance data inaccuracy has to go above 0.5 ms, which corresponds to approximately 150 km in geometric range to the satellite. Considering that the coarse time error is usually less than a few seconds, the distance error caused by it remains under a few kilometers as the maximum GNSS satellite pseudorange rate is around 800 m/s [23]. The coarse position error is also well below this error margin as the receiver can use coordinates of nearby cell towers that are typically within tens of kilometers.

#### 2.3.2. Carrier Phase Half Cycle Compensation

As mentioned in Section 2.2, the in-phase and quadrature components of the CAFs are computed for a certain data bit hypothesis. It can be observed that when a series of data bits that are exactly opposite to the actual ones is used, the resulting CAF magnitude remains the same. If we replace $H_{v}^{\left(u\right)}$ by $-H_{v}^{\left(u\right)}$ in Equation (Equation 3), the same acquisition results of code delay and Doppler offset will be found; this indicates that there are two sets of solutions with different data bit hypotheses which will meet the requirement; this is why it is named as data bit ambiguity. This can also be explained by Equation (Equation 5), the output of an arctan function is actually a series of values separated by multiples of π. That is:(9)$$\hat{\varphi }={\hat{\varphi }}_{a}+k\cdot \pi ,\;k\in Z.$$

When *k* is an even number, the integer part forms whole phase cycles and this can be assimilated to the carrier phase integer ambiguities without influencing the IAR procedure in the RTK filter, leaving ${\hat{\varphi }}_{a}$ as the only valuable phase information from the acquisition. Similarly, when *k* is an odd number, it leaves the other phase candidate of the acquisition result: ${\hat{\varphi }}_{a}+\pi$, denoted by ${\hat{\varphi }}_{b}$. This compensated half cycle distance between ${\hat{\varphi }}_{a}$ and ${\hat{\varphi }}_{b}$ implies a possible phase measurement error of half a wavelength, approximately 10 cm for GPS L1 C/A signals. This error is large enough to totally prevent the SRTK engine from achieving IAR. As a result, it is critical for open loop architecture receivers to make the correct choice between ${\hat{\varphi }}_{a}$ and ${\hat{\varphi }}_{b}$.

The data bit ambiguity issue only happens to some certain satellites at some certain epochs. This is because the actual data bits are unknown and we can only make a guess between the two hypotheses that are exactly opposite. If we can make a correct guess for all the satellites, then the resulting phase measurements will be valid and can be fed to the RTK filter for further processing. On the other way around, if all our guesses are wrong, it means all phase measurements are offset by half a cycle, and then the DD measurements will still be valid because the common offset will be cancelled out in the differencing process between satellites. The scenario that impedes the SRTK processing is when some guesses are correct while others are wrong.

In closed loop receivers, the data bit ambiguity issue can be tackled by comparing the decoded bits with a priori known bit sequences appearing within the navigation message telemetry words, as described in [24]. However, this method will not work for snapshot positioning due to the limitation of the signal duration. Fortunately, there are organizations distributing these data bit information as a service, for example the German Research Centre for Geosciences (GFZ) [22]. Once we know the exact transmission time of the signal, we can find the corresponding data bits and compare them with the bits referred by the result hypothesis. If these bits are exactly opposite, then ${\hat{\varphi }}_{b}$ should be chosen as the right carrier phase, otherwise ${\hat{\varphi }}_{a}$ should be used. As for how to obtain an exact transmission time, the next subsection provides more details.

Besides using external data sources, it must be pointed out that, for some GNSS signals with known secondary codes, the data bit ambiguity issue disappears when a longer integration time is used. As an example, Galileo E1C signal is free from such issues when the integration time used in acquisition is longer than 24 ms, equivalent to 6 primary code periods. This is because the secondary code pattern of Galileo E1C signal is known a priori; there is a maximum of 25 possible hypotheses for the received signals, with each hypothesis containing a data bit sequence of $ceil\left(T_{i}/0.004\right)+1$ digits, where $T_{i}$ denotes the integration time. If there are 8 or more bits modulated to the collected signal, it is not possible to find any two hypotheses that have exactly the same or exactly opposite bits. Thus, there is always only one data bit hypothesis output in these scenarios.

#### 2.3.3. Global Time Tag Determination

Accurate absolute time information is a cornerstone of the SRTK algorithm, it provides an edge over typical coarse time positioning engines in two ways. First, it allows satellite positions to be computed with much higher accuracy compared to those computed with a coarse time. Typically pseudorange rates of GNSS satellites vary from −800 m/s to 800 m/s, which means that the range rate difference between two satellites can reached 1600 m/s in the worst case scenarios. In those cases, even 1 ms of time error could result in 1.8 m of error in the Single Differenced (SD) measurements. Since these SD measurements are later used by the RTK filter to form Double Differenced (DD) measurements, such time errors are not bearable for high precision navigation filters. The second point where accurate timing is extremely important is for deducing the modulated data bits using external data sources. The smaller the timing error, the less possibility of retrieving the wrong bit from these distributed data set.

The absolute time accuracy in the SRTK filter has evolved through 3 stages as illustrated in the Figure 3. The first time information comes from the assistance data, typically from a cellular network, and it may include a time error of up to 2 s [14]. Then, the second stage of time accuracy is reached at millisecond level after the CTN filter generates a TOW solution based on the time from assistance data [15,25]. Finally, the third stage aims at an accuracy that is equivalent to the precision of the code delay estimates after interpolation, i.e., the corresponding range error is meter-level and the time error is then of the order of nanoseconds. The transition from stage 1 to stage 2 mainly relies on solving the five-state navigation equation, which is extensively described in GNSS literature [14,25,26], and will not be further discussed here. The key question now is how to achieve the 3rd stage of accuracy. Figure 4 presents the relationships of different time parameters that are used to answer this question.

The received signals from different satellites are transmitted at different times, as represented by the red line in Figure 4. The ultimate goal of setting a global receiver time tag is to increase the transmission time accuracy of all the satellites in view to nanosecond level. These transmission times can be expressed as:(10)$$t_{tx}^{\left(k\right)}=t_{g}-\left(X^{\left(k\right)}+\tau^{\left(k\right)}\right)=t_{g}-\tau_{a}^{\left(k\right)}-N_{sc}^{\left(k\right)}\cdot T_{sc}^{\left(k\right)},$$
where:$t_{tx}^{\left(k\right)}$ is the transmission time for satellite *k*, in seconds;$t_{g}$ represents the global time tag to be determined in this step, in seconds;$X^{\left(k\right)}$ is a multiple of secondary code periods, and can be denoted by $N_{sc}\cdot T_{sc}^{\left(k\right)}$;$N_{sc}^{\left(k\right)}$ represents an integer that also needs to be determined in this step;$T_{sc}^{\left(k\right)}$ is the duration of the secondary code for satellite *k*, in seconds;$\tau_{a}^{\left(k\right)}$ is the code delay that has been augmented by the correct navigation bit hypothesis, in seconds.

Note that Equation (Equation 10) holds only when the time tag $t_{g}$ is set at the edge of a secondary code period in a GNSS time scale, otherwise the integer nature of $X^{\left(k\right)}$ is no longer maintained. In addition, $\tau_{a}^{\left(k\right)}$ is a modulo-$T_{sc}^{\left(k\right)}$ value and can be computed using
(11)$$\tau_{a}^{\left(k\right)}={\hat{\tau }}^{\left(k\right)}+N_{c}^{\left(k\right)}\cdot T^{\left(k\right)},$$
with the code delay ${\hat{\tau }}^{\left(k\right)}$ and the $N_{c}^{\left(k\right)}$ value previously estimated using Equation (Equation 8).

The next step is to determine $t_{g}$ and the corresponding number of secondary code cycles to be added for each satellite, i.e., the $N_{sc}^{\left(k\right)}$ value, while maintaining the relative code delays relationship between satellites. In principle, the time tag can be chosen at any secondary code edge, but since the CTN filter has provided a coarse time estimate $t_{ct}$ with millisecond level accuracy, it will be used as an entry point and the time tag will be computed in the following three sub-steps. The first sub-step is to transfer the millisecond level timing accuracy from the TOW solution inside the CTN filter output to the satellite transmission time based on Equation (Equation 12):(12)$$t_{tx\_rough}^{\left(k\right)}=t_{ct}-\frac{\rho_{a}^{\left(k\right)}}{c},$$
where $\rho_{a}^{\left(k\right)}$ is the geometric range computed with the assistance data. As already mentioned, any assistance data inaccuracy will impact the signal’s flight time for less than half a millisecond; thus, it can be safely used for the computation of the rough transmission time while still maintaining a millisecond level accuracy. Ideally, by adding the fractional code delay to the actual satellite transmission time should bring us to an epoch at the edge of a secondary code period. Thus, it can be expected that the term $t_{tx\_rough}^{\left(k\right)}+\tau_{a}^{\left(k\right)}$ will also be near a secondary code edge with a millisecond-level error. Then, we can choose the satellite with the largest secondary code period, denoted by $T_{sc}^{max}$, as the reference satellite, and set the global time tag as
(13)$$t_{g}=\mathrm{round}\left(\frac{t_{tx\_rough}^{\left(k\right)}+\tau_{a}^{\left(k\right)}}{T_{sc}^{max}}\right)\cdot T_{sc}^{max}.$$

Finally, the corresponding secondary code integer value $N_{sc}^{\left(k\right)}$ for satellite *k* can be computed as
(14)$$N_{sc}^{\left(k\right)}=\mathrm{round}\left(\frac{t_{g}-t_{tx\_rough}^{\left(k\right)}-\tau_{a}^{\left(k\right)}}{T_{sc}^{\left(k\right)}}\right).$$

It must be pointed out that the global time tag and the integer number of secondary code periods come as a pair. When one value changes, the other one also need to be modified correspondingly in order not to impact the satellite transmission time accuracy. The final pseudorange measurements corresponding to the receiver time tag $t_{g}$ can be computed by
(15)$$P_{final}^{\left(k\right)}=c\cdot \left(\tau_{a}^{\left(k\right)}+N_{sc}^{\left(k\right)}\cdot T_{sc}^{\left(k\right)}\right).$$

Although this method should work for most scenarios, it may still be prone to errors when the TOW solutions generated in the CTN filter contain an error larger than half the period of the maximum secondary code. That can happen, for example, when the Dilution Of Precision (DOP) is really high. As an example, the secondary code period for Galileo E1C signal is 100 ms. If this signal is used in the SRTK filter, as in our experiments, the coarse time TOW solution error should not exceed 50 ms, even though it is very unlikely [25], otherwise the global time tag determination method explained above could fail to produce satellite transmission times with nanosecond-level accuracy.

### 2.4. SRTK Performance

This section introduces the RTK measurement model and the method used to fix the carrier phase ambiguities in a single epoch. Then the parameters of interest for this work are discussed as well as their impact on SRTK performance.

#### 2.4.1. RTK Filter

There are usually different algorithm configurations for RTK depending on the baseline distance. For long baseline scenarios, typically the following DD measurements models are used [27].
(16)$$\begin{array}{c} \nabla \Delta P_{i}=\nabla \Delta \rho +\nabla \Delta I+\nabla \Delta T+\nabla \Delta \varepsilon_{i,P},\\ \nabla \Delta \lambda_{i}\phi_{i}=\nabla \Delta \rho +\lambda_{i}\nabla \Delta N_{i}-\nabla \Delta I+\nabla \Delta T+\nabla \Delta \varepsilon_{i,\phi }, \end{array}$$
where:∇Δ is the DD operator;*P* and ϕ are the pseudorange in the m and carrier phase measurement in cycles, respectively;$\lambda_{i}$ is the signal wavelength for carrier frequency *i*, in m;*T* and *I* are the Troposphere and Ionosphere delay, respectively, in m;ε represents the measurement noise, in m;*N* stands for the carrier phase integer ambiguity.

In these cases, the vector of unknowns includes the parameters for Ionosphere and Troposphere delays. For the Ionosphere delay, this parameter can be the Total Electron Content (TEC). Moreover, a variant strategy is to build the Ionosphere Free linear combination of measurements from different carrier frequencies [28], however, this is not applicable to the present research as our focus in on single frequency snapshots. As for Troposphere delay, it can be estimated as a Zenith Total Delay (ZTD) and the North and East components of the Troposphere gradient [29].

However, for short baseline cases, the Ionosphere and Troposphere delays and other minor correction terms such as the phase center variations are assumed to be eliminated through the DD operation [30]. As a result, the models presented in Equation (Equation 16) can be simplified as follows:
(17)$$\begin{array}{c} \nabla \Delta P_{i}=\nabla \Delta \rho +\nabla \Delta \varepsilon_{i,P},\\ \nabla \Delta \lambda_{i}\phi_{i}=\nabla \Delta \rho +\lambda_{i}\nabla \Delta N_{i}+\nabla \Delta \varepsilon_{i,\phi }. \end{array}$$

In this research, only the short baseline model is considered for simplicity and also due to the fact that only single frequency snapshots are considered. This means that the residual DD Ionosphere and Troposphere delays that are actually not fully cancelled out are lumped into the DD measurement noise term ∇Δε; thus, it can be expected that the RTK performance will degrade when the baseline grows larger.

After the float solutions are generated using these measurements, the Least-squares Ambiguity Decorrelation Adjustment (LAMBDA) method is applied for the IAR procedure [31]. Due to the nature of snapshot positioning, only a single epoch of data is available. Thus, these measurements are processed in instantaneous mode. To determine if the current ambiguities are fixed, the distance between the best integer estimates to the float solution is calculated. This process is repeated for the second best integer set and then the ratio between these two distances are computed, resulting in the so called LAMBDA Ratio Factor (LRF). By comparing it to a defined threshold value, we can decide whether the fix is successful. Note that this may not be the optimal acceptance test in terms of failure rate or probability of producing a wrong fix [32], but is the simplest to implement. In this research, the RTK performance is mainly defined by the mean LRF value and the fix rate, i.e., the percentage of epochs that have a LRF value higher than a certain threshold.

#### 2.4.2. Parameters of Interest

The relationships between the parameters of interest are presented in Figure 5. The main purpose of this work is to explore the impact of the following 3 parameters on the SRTK performance: snapshot signal bandwidth, integration time, and baseline distance, which are shaded in red color in Figure 5.

Starting from these 3 parameters, we can explore on the left and right directions to find their impacts on the SRTK fix rate and the snapshot data size, respectively.

Looking leftwards, we can see first of all that the observables noise level of the rover is determined by the signal bandwidth and the integration time [2,33,34], while the baseline distance impacts the magnitude of the atmosphere delay variation between the rover and the base. The rover observable noise terms can be evaluated by their Cramér-Rao Bounds (CRBs), which are proportional to $\propto \frac{1}{\left(C/N_{0}\right)BT_{\mathrm{int}}}$ and $\propto \frac{1}{\left(C/N_{0}\right)T_{\mathrm{int}}}$ for, respectively, the code delay and carrier phase observables [11,35]. Here the $T_{\mathrm{int}}$, *B* and $C/N_{0}$ variables denote the integration time, signal bandwidth and carrier-to-noise-density-ratio, respectively. It can be seen that both code delay and carrier phase errors decreases when the integration time $T_{\mathrm{int}}$ increases, while the bandwidth *B* only influences the code delay. The rover observable noise, together with other error terms such as reference receiver measurement noise and satellite position errors cause by timing inaccuracies are added up to form the DD measurement errors. Note that the atmosphere delay variations are also lumped into such errors since a simple DD measurement model is used in this research. The magnitude of the DD measurement errors directly decide the LRF value and the final fix rate. The RTK algorithm configuration also plays a role in the computation of fix rate but, since it is out of scope of this work, we applied the same configuration for all the experiments performed to disregard its impact.

Looking rightwards, the bandwidth puts a constraint on the signal sampling rate in order to avoid the signal aliasing problems, that is, we must set the sampling rate no lower than the Nyquist rate, which is usually twice the bandwidth [36]. On the other hand, the integration time sets a minimum value of the total duration of the captured signal. These two parameters, together with the quantization factor, directly determine the snapshot data size with:
(18)S=2·Q×Fs×T+A, where:
*S* is the size of the snapshot data, in bits;*Q* represtns the constant quantization parameter, in bits per sample;Fs stands for the sampling rate, in Hz;*T* is the total integration time, in s;*A* stands for the size of additional data, in bits.

The additional data size term *A* is usually very small (less than 1 kB) since only very limited amount of metadata is needed along side the snapshot signal data bits, such as rover receiver ID. The multiplication by the constant 2 is due to the I and Q streams of the snapshot signal.

As can be seen, changes to the 3 main parameters influence both the fix rate and the snapshot data size at the same time. The following experiments further explore the relationships between these parameters of interest.

## 3. Experiment Setup and Results

### 3.1. Data Collection

A set of snapshots were recorded for this research using a snapshot receiver designed by Albora Technologies. A high-gain antenna (Septentrio PolaNt-x MF [37]) was used and located in an open environment at the north campus of Universitat Politècnica de Catalunya (UPC) in Barcelona, as shown in Figure 6.

The collected snapshot data were sent to the Albora cloud platform and were then processed by the SRTK positioning engine proposed in this manuscript. The base station data were collected using the Virtual Reference Station (VRS) service provided by Institut Cartogràfic i Geològic de Catalunya (ICGC). Several streams of VRS data were collected simultaneously by setting a series of reference points that have a different distance to the snapshot receiver; in this way, reference data for different baseline distances were collected. The GPS navigation data bits were collected from the GFZ server to address data bit ambiguity issue and compensate the half cycle phase biases properly. A total of 240 snapshots were collected over 4 h, with each snapshot datum separated by one minute. The original data contain signals from the L1 and G1 bands; however, only L1 signals were used for this research. The default front-end sampling rate was 63.6 MHz and each snapshot had an original duration of 100 ms.

### 3.2. Experiment Setup

In order to explore different performances while using different parameters of interest, the following configuration choices were made, and the snapshot data set was processed using all combinations of these settings:Integration time (ms): {40,60,80,100};Signal bandwidth (MHz): {6.36,7.95,10.6,12.72,15.9,21.2,25.44,31.8};Baseline distance (km): {0,5,10,15,20,25,30,35,40,45,50}.

As mentioned before, the integration time was controlled by cropping the original collected signal to the desired duration and the signal bandwidth was adjusted in the front-end by low-pass filtering and then resampling to the Nyquist frequency. The baseline distance change was implemented by feeding different VRS data files to the RTK filter. The RTK positioning engine used for this research is RTKLIB, [30,38]. The RTK fix rate was finally calculated with the LRF threshold set to two.

### 3.3. Results and Discussions

Before presenting the RTK performances, the number of satellites and DOP values are shown in Figure 7 as an overview of the snapshot data scenario geometry, an elevation mask of 10 degrees was set to filter out the low-elevation satellites.

Note that this work focuses on single frequency applications and only GPS L1 C/A and Galileo E1C signals wereused for the following RTK process. Although BDS B1C signals are also in the L1 frequency band; their measurements were not provided by the VRS service used in this research at the time of data collection, and instead only the BDS B1I signal which is centered at a slightly different frequency (1561.098 MHz) was available, thus the BDS constellation is not included in the current SRTK engine analysis.

To present the fix rate and mean LRF results, two plots can be generated for each integration time used, with the X axis representing the varying settings of baseline distance and Y axis representing the SRTK fix rate and mean LRF value, respectively. As an example, Figure 8 and Figure 9 are generated for all the bandwidths used when the integration time is set to 100 ms.

As expected, the two plots follow a similar pattern, since the LRF directly impacts the possibility of obtaining a RTK fix. For other integration time settings, similar trends can be found in the results.

#### 3.3.1. Baseline Distance Impact

By analyzing Figure 8 and Figure 9, it can be clearly observed that generally both the mean LRF value and SRTK fix rate drop when the baseline distance increases. When the baseline distance is less than 10 km, this trend does not appear to be very obvious as the SRTK fix rate curve is almost flat and the mean AR ratio variation is also small. When we extend the baseline further, the SRTK fix rate drops more rapidly, which could be caused by the fact that the simple DD measurement model adopted in this research is meant for short baseline scenarios. With 100 ms of integration time, and a baseline shorter than 15 km, more than 90% of snapshots generated valid RTK fixed solutions, regardless of the signal bandwidths been tested. When the baseline was extended to 50 km, the fix rate dropped to approximately 30%, with an average LRF value below 2.

#### 3.3.2. Signal Bandwidth Impact

As shown in Figure 8, the bandwidth change has some impact on the fix rate as well. The green line representing 6.36 MHz of sampling rate is lower than the one for 31.8 MHz by approximately 5% for most baseline distance cases. However, this impact seems much less significant compared to influences of baseline distance. As also shown in Figure 9, the mean LRF value only slightly decreased from 7.2 to 5.8 for short baselines. This change is even less noticeable under longer baseline scenarios.

#### 3.3.3. Integration Time Impact

The fix rate and mean LRF results for different integration times at different baselines are collected in Table 1 and Table 2, where a signal bandwidth of 31.8 MHz was used to generate these results.

As expected, both tables show that the performance of longer integration time is generally better compared to those shorter ones, although its impact is still not as significant as the baseline distance. By looking at Table 1, an 8% improvement is observed when increasing the integration time from 40 ms to 100 ms, while this improvement becomes less noticeable when the baseline distance grows larger than 40 km. This table also shows that the fix rate can still reach 90% for all the integration times used when the baseline distance is smaller than 10 km, and a fix rate of 80% can be assured for 20 km baselines. Results in Table 2 further confirm that the mean LRF result improves when a longer integration time is used. An overall 20% improvement in the mean LRF value can be seen when integration time increases from 40 ms to 100 ms, except for the scenarios with baseline distances over 40 km.

#### 3.3.4. Snapshot Size

As represented by Equation (Equation 18), the snapshot data size depends linearly on the sampling rate and signal duration. The data sizes corresponding to the chosen settings are computed and represented in Table 3 together with the resulting SRTK fix rates when these settings are used.

To show the relationship between the SRTK fix rate and the data size, all their values at 15 km baseline distance have been presented in Figure 10. Note that results from different integration times are represented by different colored markers to show that snapshot data size increases along with the integration time. In general, larger snapshot sizes result in a higher possibility of fixing SRTK ambiguities. However, also notice that the smallest data size used in this research is 127.2 kB and results in a 78.84% fix rate, while the largest data size used is 1590 kB, which is 12.5 times the smallest data size, and brings a fix rate of 96.27%. There is an obvious trade-off to consider between the data size and the expected accuracy. Assuming a minimum acceptable fix rate of 90%, the smallest data size able to achieve this goal in the test scenario is 254.4 kB, by using 12.72 MHz of bandwidth and 40 ms of integration time, for an observed fix rate of 93.36%.

#### 3.3.5. Positioning Accuracy

The output coordinate errors are plotted in Figure 11 and Figure 12 to show the accuracy performance of SRTK when carrier phase ambiguities are fixed. These results are generated using 100 ms of integration time and 31.8 MHz signal bandwidth under 5 km baseline distance.

Figure 11 shows a small common bias in the solutions, respectively, −0.11 cm, 0.93 cm and 0.48 cm for East, North and Up directions. These biases might be a result of the inaccuracies in the ground truth, which was computed by averaging RTK-fixed solutions from a closed-loop receiver. The corresponding standard deviations are 0.29 cm, 0.41 cm and 0.59 cm. These results prove that centimeter level accuracy can be achieved with just a brief interval of signal collected by a GNSS front-end.

Finally, the horizontal errors are also presented in Figure 12; as can be seen, the position points are generally located inside a 2 cm × 2 cm region with a calculated Circular Error Probable (CEP) of 1.012 cm, as denoted by the red circle. The Root Mean Square (RMS) horizontal error is 1.066 cm and the 3D-RMS error is 1.309 cm. The 95-percentile horizontal (denoted by the green circle) and 3D errors were 1.619 cm and 1.921 cm, respectively.

## 4. Conclusions

The results presented in this paper show how SRTK is able to achieve IAR and obtain a high accuracy solution with long baselines of up to 50 km, confirming for the first time that the SRTK technique can be practically functional in real scenarios, i.e., non-zero baselines. The manuscript explored the influences of signal bandwidth, integration time and baseline distance, and for this purpose, the collected snapshot data set was processed using different combinations of settings. The results show that higher bandwidths and longer integration times lead to better performances of the SRTK engine in terms of RTK fix rate and mean LRF value for all the snapshots, although at the price of a larger size of data to be transmitted and a higher computational burden. These two are the dominant impacting parameters when the distance is shorter than 10 km. However, it is also noticed that the baseline distance has a more significant impact on the fix rate than the bandwidth and integration time when the distance is larger than 10 km, which could be caused by the simple DD measurement model we applied for the navigation filter. In addition, it was recorded that, for baselines shorter than 15 km, a fix rate of 90% was reached for all the settings combinations. The test results also show that a snapshot with the size of 255 kB could result in a potential fix rate of more than 93%, with centimeter level positioning accuracy when carrier phase ambiguities are correctly fixed.

However, there is still a lot of unexplored potential in SRTK and better performance could be achieved with some further research. First of all, this work has a limited the number of satellites by using only GPS and Galileo constellations. With a more complete infrastructure network, more satellites from other constellations can be included in the RTK algorithm and higher fix rates are expected. Secondly, the ratio test could be implemented using LRF threshold values that adapt to a given failure rate, instead of directly comparing with the fixed value of two that is used in this research, the change of strategy could bring a big improvement in fix rate especially for the cases where the LRF values are close to the selected threshold. Thirdly, a more robust method can be developed for GPS satellites to solve the data bit ambiguity issue, the goal would be to remove the dependence on external navigation data bit services. Finally, more detailed performance evaluations about IAR and fix rate can be done by comparing the results with ground truth values derived based on a known baseline.

## Figures and Tables

**Figure 1 sensors-21-03688-f001:**
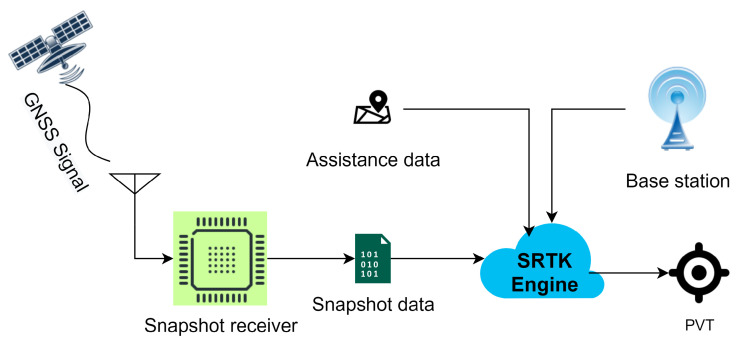
General workflow of cloud-based GNSS positioning.

**Figure 2 sensors-21-03688-f002:**
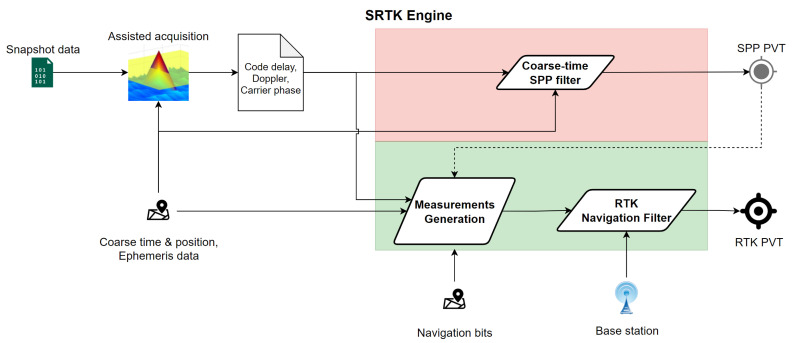
SRTK data processing steps.

**Figure 3 sensors-21-03688-f003:**
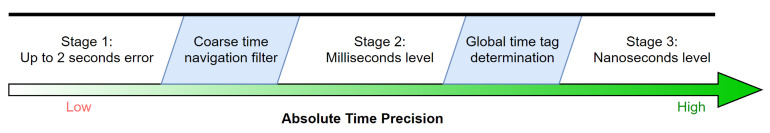
Time precision evolution in the SRTK positioning engine.

**Figure 4 sensors-21-03688-f004:**
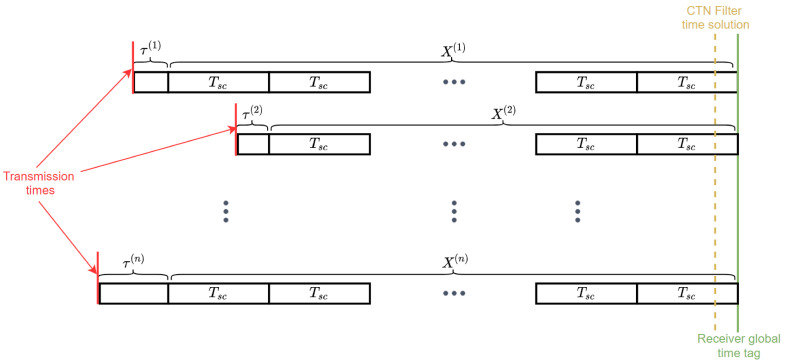
Relationship between the global time tag and other time parameters.

**Figure 5 sensors-21-03688-f005:**
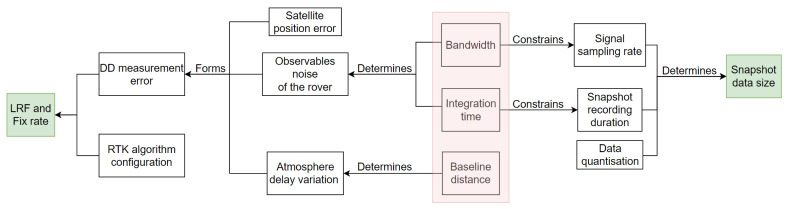
Relationships between the parameters of interest.

**Figure 6 sensors-21-03688-f006:**
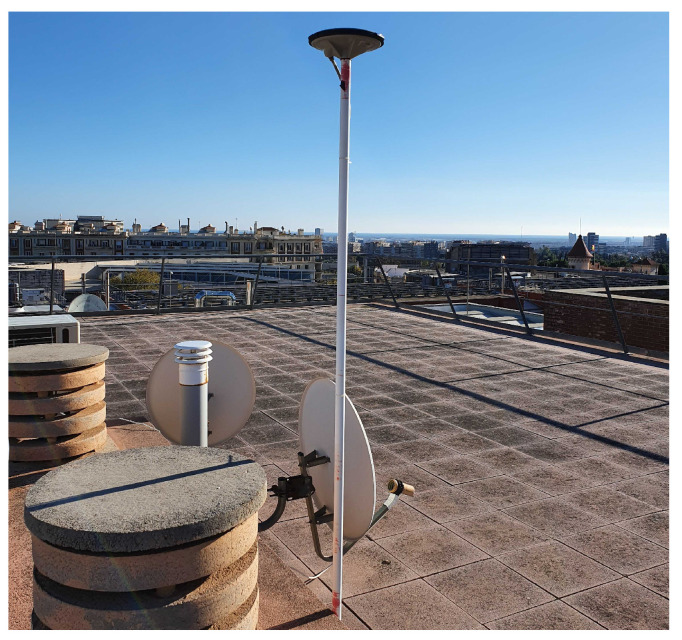
Antenna location for snapshot data collection.

**Figure 7 sensors-21-03688-f007:**
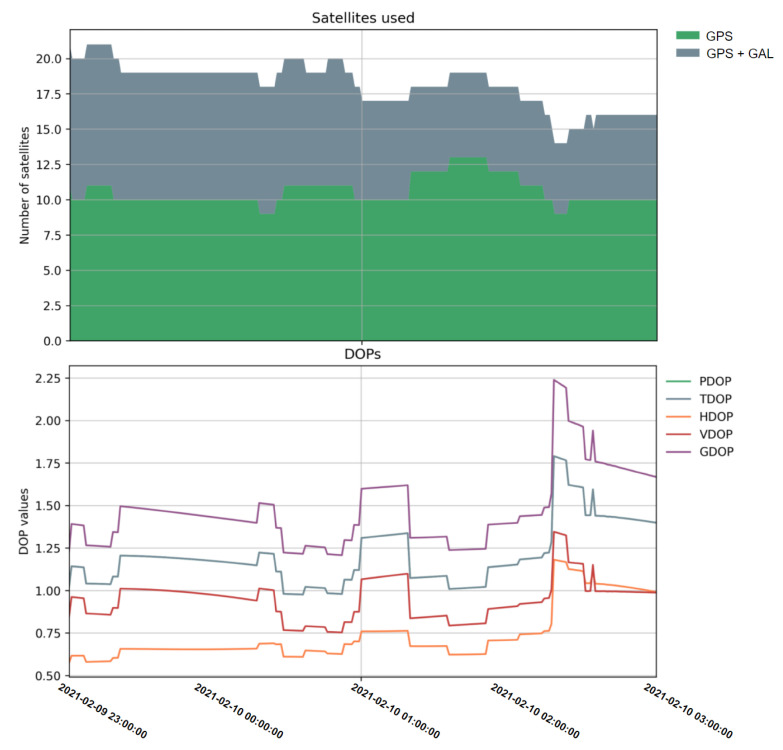
Number of satellites used (**top**) and the DOP values (**bottom**) of the collected snapshot data.

**Figure 8 sensors-21-03688-f008:**
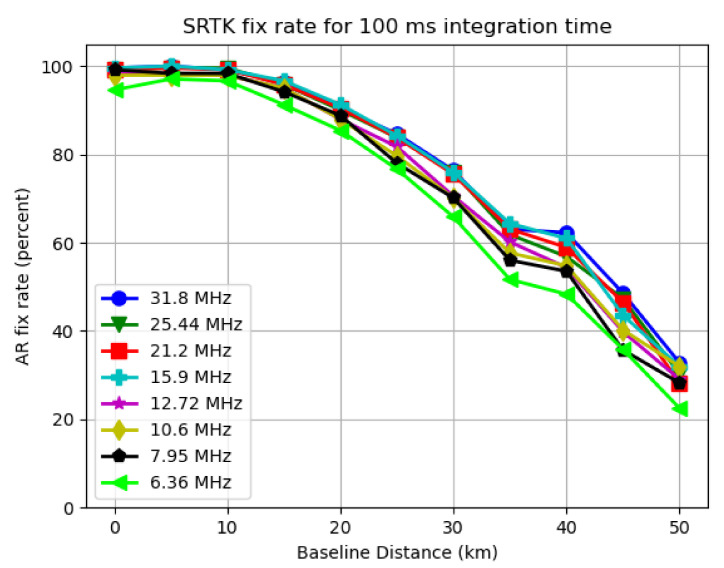
SRTK fix rate for different bandwidths when 100 ms integration time is used.

**Figure 9 sensors-21-03688-f009:**
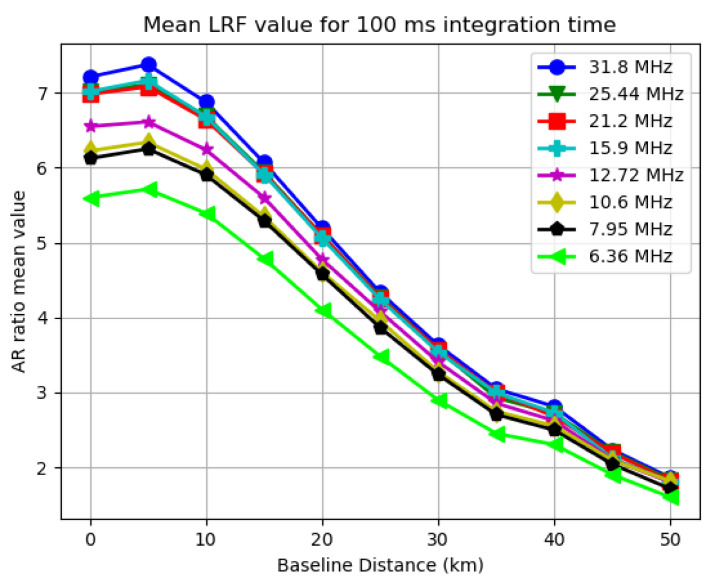
Mean LRF value for different bandwidths when 100 ms integration time is used.

**Figure 10 sensors-21-03688-f010:**
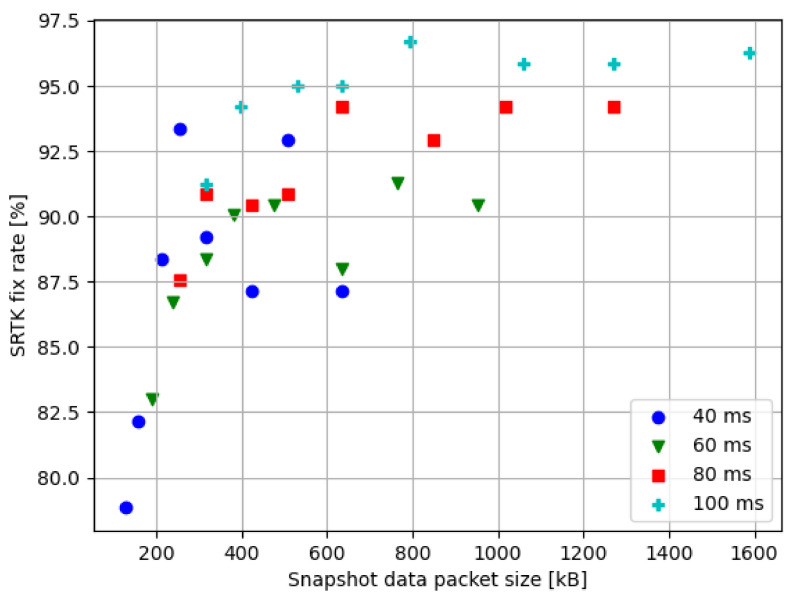
SRTK fix rates in relation to the snapshot sizes at 15 km baseline.

**Figure 11 sensors-21-03688-f011:**
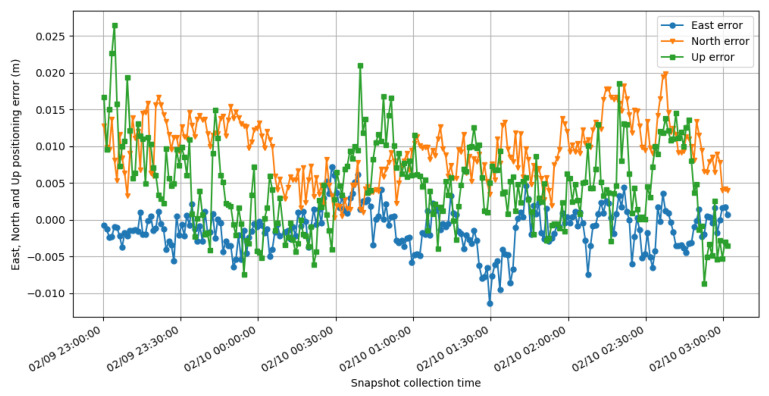
East (blue), North (orange) and Up (green) errors of SRTK fixed solutions under 5 km baseline.

**Figure 12 sensors-21-03688-f012:**
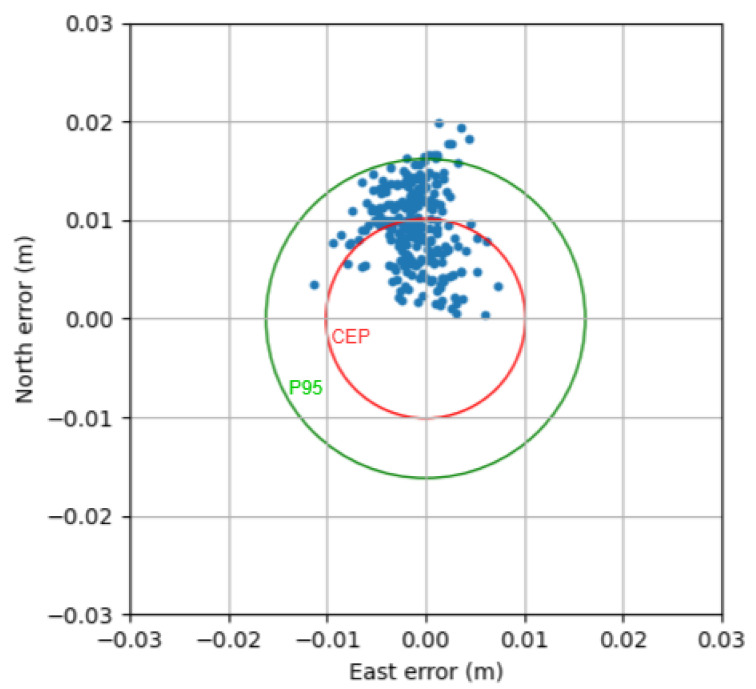
Horizontal positioning errors in East and North under a 5 km baseline.

**Table 1 sensors-21-03688-t001:** RTK fix rate (in percentage) for different integration times at different baselines.

	5 km	10 km	15 km	20 km	25 km	30 km	35 km	40 km	45 km	50 km
**40 ms**	92.53	90.87	87.14	82.16	74.69	66.39	53.11	51.45	40.66	31.54
**60 ms**	95.44	95.02	90.46	84.65	78.01	70.54	55.6	46.89	37.34	28.63
**80 ms**	98.34	98.76	94.19	88.38	80.91	74.69	59.34	56.02	43.98	27.8
**100 ms**	100	99.17	96.27	90.46	84.65	76.35	63.07	62.24	48.55	32.78

**Table 2 sensors-21-03688-t002:** Mean LRF values for different integration times at different baselines.

	5 km	10 km	15 km	20 km	25 km	30 km	35 km	40 km	45 km	50 km
**40 ms**	5.92	5.56	4.99	4.35	3.69	3.06	2.51	2.38	2.03	1.73
**60 ms**	6.58	6.18	5.49	4.67	3.9	3.23	2.64	2.32	2.04	1.73
**80 ms**	7.15	6.72	5.93	4.99	4.1	3.48	2.81	2.55	2.14	1.74
**100 ms**	7.38	6.88	6.07	5.19	4.33	3.63	3.05	2.81	2.23	1.86

**Table 3 sensors-21-03688-t003:** Snapshot data size (1st value, in kB) and SRTK fix rate (2nd value, percentage) for different integration times and bandwidths at 15 km baseline.

	31.8 MHz	25.44 MHz	21.2 MHz	15.9 MHz	12.72 MHz	10.6 MHz	7.95 MHz	6.36 MHz
**40 ms**	636	508.8	424	318	254.4	212	159	127.2
	87.14	92.95	87.14	89.21	93.36	88.38	82.16	78.84
**60 ms**	954	763.2	636	477	381.6	318	238.5	190.8
	90.46	91.29	87.97	90.46	90.04	88.38	86.72	82.99
**80 ms**	1272	1017.6	848	636	508.8	424	318	254.4
	94.19	94.19	92.95	94.19	90.87	90.46	90.87	87.55
**100 ms**	1590	1272	1060	795	636	530	397.5	318
	96.27	95.85	95.85	96.68	95.02	95.02	94.19	91.25

## Data Availability

The study used data from ICGC and GFZ for generating the results. The GFZ data can be accessed through the URL http://isdc.gfz-potsdam.de (accessed on 25 May 2021), project: GNSS, product type: GNSS-GPS-1-NAVBIT. The ICGC data can be obtained at ftp://geofons.icgc.cat/rinex3/ (accessed on 25 May 2021).

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
