# Peer review of "Cloud-Based Single-Frequency Snapshot RTK Positioning"

_sensors, 2021, doi:10.3390/s21113688_

Round 1

Reviewer 1 Report

This paper presents a study on the cloud-based snapshot RTK positioning (SRTK). The workflow of the SRTK was introduced, and the challenges of achieving an SRTK fix were discussed from the perspective of signal bandwidth, integration time and baseline distance. I think this is a very interesting study, but the author is rather rough in the analysis and discussion of the results. It can’t be published in current form, and there are some questions needing clarifications as following.

  1. In the section of “Methodology”, the signal acquisition and observation generation of SRTK were introduced. Is there any innovation or improvement in the method used here? And, what is the difference between the GNSS data used in SRTK and traditional RTK?
  1. Line 328, page 10: What is the difference of RTK filtering processing strategy in different baseline distance? For example, the ionosphere errors and troposphere errors. It seems that the difference in positioning accuracy in this paper is mainly caused by the baseline distance. And this conclusion is already recognized, so what is the main contribution of this paper?
  1. Figure 7 and 8, page 13: In this analysis, the GNSS observation quality of different signal bandwidth should be further evaluated. Since positioning accuracy is also affected by RTK algorithm configuration, more attention should be paid to data quality itself. I suggest to supplement the relevant results and analysis here.
  1. Table 3, page 15: In this table, the effect of snapshot size on positioning performance does not seem to be found. Why does the snapshot size here affect the fixed rate?
  1. It was suggest that the author first analyze the influencing factors of the data quality of the rover station in SRTK, and then analyze the positioning performance of the data under different circumstances.

Reviewer 2 Report

The present paper introduces a cloud-based Snapshot Real Time Kinematics and evaluates the impact of parameters such as signal bandwidth, integration time, and baseline distances in the system performance. The paper also assesses the SRTK performance in terms of Integer Ambiguity Resolution and positioning accuracy.

The paper concerns an interesting topic in the precise GNSS positioning field. The organization and the presentation of the paper are good. The novelty is fair since the proposed work comes from previous works; however, the system performance evaluation merits. Below, some minor comments the author should address:

  • “This work also describes a novel solution to ensure a nanosecond level absolute timing accuracy in order to compute highly precise satellite coordinates”. This aspect is omitted in the introduction;
  • In the introduction, the contribution of the paper is spread along the entire section. Please clarify the contribution of the paper, also by using bullet points if necessary;
  • Please also provide prospective drawbacks of cloud-based snapshot positioning;
  • “The resulting output file is the snapshot data that will be transmitted to the cloud platform by the cellular network module.” Is the snapshot data containing the digitized data transmitted from the satellites? If so, this is a large quantity of information. Is the author sure that the transmission of this information will lead to save energy?
  • Please add a reference about coarse-time navigation between 2009 and 2021 (e.g. Peer‐to‐peer cooperation for GPS positioning)
  • “The latter is less commonly used but is more practical for snapshot”. As for the closed-loop, briefly describes the open-loop approach.
  • “snapshot receiver front-end designed by Albora Technologies.”. Can the author provide more information about this front-end?
  • “L1 and G1 bands”. G1 or E1?
  • “Baseline Distance Impact”. Please add a brief discussion regarding the potential reasons why the performance decreases with the baseline distance. The same for the signal bandwidth and the integration time.

Reviewer 3 Report

Very appealing paper reporting valid findings. Furthermore, it addresses a currently hot topic in applications. The reading looks not so immediate, but overall I guess that readers (maybe better exoert readers) would understand it in full. Just a couple of issues:

- line 139: The "normalization by sampling rate" is not fully clear to me (and maybe to future readers too);

- line 182: the half a cycle bias for some satellites maybe needs a bit of explanation;

- line 246, it could be interesting to better detail which is the type/amount of data required from these services. It looks that it is related to the duration of the snapshot plus  an additional "cushion". And, it would simply consist in the indications of the correct binary value for the time interval of interest. Maybe a practical example could help.

- the final claim about RAIM is not clearly motivated, and actually not related to the content of the manuscript.

Round 2

Reviewer 1 Report

The manuscript presents a very interesting study about RTK. And it has been substantially revised and significantly improved. I am basically satisfied with the efforts by the authors.

Author Response

The authors would like to thank the reviewer for his insightful comments that helped to improve the manuscript.